# Spontaneous Right Ventricular Pseudoaneurysms and Increased Arrhythmogenicity in a Mouse Model of Marfan Syndrome

**DOI:** 10.3390/ijms21197024

**Published:** 2020-09-24

**Authors:** Felke Steijns, Marjolijn Renard, Marine Vanhomwegen, Petra Vermassen, Jana Desloovere, Robrecht Raedt, Lars E. Larsen, Máté I. Tóth, Julie De Backer, Patrick Sips

**Affiliations:** 1Center for Medical Genetics, Department of Biomolecular Medicine, Ghent University, 9000 Ghent, Belgium; Felke.Steijns@UGent.be (F.S.); Marjolijn.Renard@UGent.be (M.R.); Marine.Vanhomwegen@UGent.be (M.V.); Petra.Vermassen@UGent.be (P.V.); Julie.DeBacker@UGent.be (J.D.B.); 24BRAIN, Department of Head and Skin, Ghent University, 9000 Ghent, Belgium; Jana.Desloovere@UGent.be (J.D.); Robrecht.Raedt@UGent.be (R.R.); LarsEmil.Larsen@UGent.be (L.E.L.); 3Institute Biomedical Technology, Ghent University, 9000 Ghent, Belgium; MateIstvan.Toth@UGent.be; 4Department of Cardiology, Ghent University Hospital, 9000 Ghent, Belgium

**Keywords:** marfan syndrome, fibrillin-1, cardiac function, myocardial compaction, ventricular pseudoaneurysm, electrocardiogram, arrhythmia

## Abstract

Patients with Marfan syndrome (MFS), a connective tissue disorder caused by pathogenic variants in the gene encoding the extracellular matrix protein fibrillin-1, have an increased prevalence of primary cardiomyopathy, arrhythmias, and sudden cardiac death. We have performed an in-depth in vivo and ex vivo study of the cardiac phenotype of *Fbn1^mgR/mgR^* mice, an established mouse model of MFS with a severely reduced expression of fibrillin-1. Using ultrasound measurements, we confirmed the presence of aortic dilatation and observed cardiac diastolic dysfunction in male *Fbn1^mgR/mgR^* mice. Upon post-mortem examination, we discovered that the mutant mice consistently presented myocardial lesions at the level of the right ventricular free wall, which we characterized as spontaneous pseudoaneurysms. Histological investigation demonstrated a decrease in myocardial compaction in the MFS mouse model. Furthermore, continuous 24 h electrocardiographic analysis showed a decreased heart rate variability and an increased prevalence of extrasystolic arrhythmic events in *Fbn1^mgR/mgR^* mice compared to wild-type littermates. Taken together, in this paper we document a previously unreported cardiac phenotype in the *Fbn1^mgR/mgR^* MFS mouse model and provide a detailed characterization of the cardiac dysfunction and rhythm disorders which are caused by fibrillin-1 deficiency. These findings highlight the wide spectrum of cardiac manifestations of MFS, which might have implications for patient care.

## 1. Introduction

The myocardial extracellular matrix (ECM) is a complex fibrous meshwork in close contact with cardiac myocytes, fibroblasts, leukocytes, and endothelial cells [1], which plays an important role in myocardial structure and function, thereby continuously adapting to local cell signaling. Together with the fibrillar proteins collagen, elastin, and fibronectin, fibrillin-1 is an important component of the myocardial ECM and is believed to be involved in the transmission of forces from the myocardial ECM to the cardiac myocytes, based on its specific spatial arrangement in the myocardial tissue [2,3,4]. Pathogenic *FBN1* variants resulting in defective fibrillin-1 cause Marfan syndrome (MFS). MFS is an autosomal dominantly inherited connective tissue disorder associated with life threatening cardiovascular manifestations, of which aortic dilatation/dissection and mitral valve prolapse are major causes of morbidity and mortality [5,6,7].

Better clinical management has led to an increased life expectancy for patients with MFS in recent decades. Nevertheless, combined with improved cardiovascular surveillance, this has revealed additional MFS-related cardiovascular problems, such as intrinsic myocardial dysfunction and arrhythmia [8,9,10,11,12,13,14,15,16]. In order to explore these MFS-related cardiac manifestations, several studies have already been performed in *Fbn1^C1039G/+^* and *Fbn1^mgR/mgR^* MFS mouse models [17]. The *Fbn1^C1039G/+^* mouse model is an antimorphic model resulting from the substitution of a conserved cysteine to a glycine in a calcium-binding EGF-like domain of *Fbn1*. Heterozygous *Fbn1^C1039G/+^* mice show a mild, classic MFS phenotype. The *Fbn1^mgR/mgR^* mouse model, on the other hand, is a hypomorphic model. Due to the insertion of a neomycine cassette between introns 18 and 19 of *Fbn1*, fibrillin-1 expression is reduced approximately five-fold. Homozygous *Fbn1^mgR/mgR^* mice show an early-onset, severe MFS phenotype.

Cua et al. [18] and Lee et al. [19] demonstrated a significant dilatation of the left ventricle (LV) in the *Fbn1^C1039G/+^* mouse model by means of optical coherence tomography. Mild LV dilatation, independent of valvular dysfunction, was also observed by Campens et al. [11] in the same mouse model using ultrasound analysis. In other studies by Cavanaugh et al. [20] and Rouf et al. [21], a significant increase in the LV diameter of *Fbn1^C1039G/+^* mice was only observed after a two-week angiotensin II treatment, or after surgical constriction of the thoracic aorta, respectively. Cook et al. [22], on the other hand, identified a severe primary dilated cardiomyopathy with significant systolic dysfunction in the *Fbn1^mgR/mgR^* mouse model. They also found a significant broadening of the QRS complex on the electrocardiogram (ECG) of *Fbn1^mgR/mgR^* mice in comparison to their wild-type littermates, suggestive of ventricular conduction delay.

Currently, the number of mouse studies regarding MFS-related myocardial dysfunction and arrhythmia is limited. Additional research on the cardiac phenotype of MFS mouse models is warranted as this might lead to novel insights in the underlying pathophysiological processes, which might have implications for the management of cardiac manifestation of MFS in patients. In this study, we therefore performed in-depth cardiac phenotyping analyses of the *Fbn1^mgR/mgR^* mouse model, which included a detailed in vivo and ex vivo evaluation of cardiac structure and function.

## 2. Results

### 2.1. Ultrasound Analysis of Thoracic Aorta Diameter and Cardiac Function

A marked dilatation was observed at all levels of the thoracic aorta in the *Fbn1^mgR/mgR^* (MFS) mice compared to wild-type (WT) mice, as a hallmark of the MFS phenotype (Table 1). To investigate cardiac valve function, ultrasound analysis of intracardiac flow patterns was performed. Aortic- and/or mitral valve regurgitation (presenting as holodiastolic flow reversal at the level of the aortic valve and the mitral valve, respectively) was observed in 73% of the MFS mice. None of the WT mice presented significant valve regurgitation.

Cardiac function was also calculated from ultrasound recordings. No significant difference was found for the LV end-diastolic and end-systolic diameter (LVEDD and LVESD, respectively) between the WT and MFS mice. In addition, the right ventricular end-diastolic diameter (RVEDD) and LV fractional shortening (FS) revealed no significant difference between the MFS mice and their WT littermates. However, a significant decrease of E wave velocity (early filling of the left ventricle) could be observed in the MFS mice compared to their WT littermates (independent samples test; *p* = 0.002). As no difference could be observed in the values of the A wave velocity (late filling of the left ventricle) between the MFS and WT mice, this decrease in E wave peak amplitude resulted in a decreased E/A ratio in the MFS mice (Table 1).

### 2.2. Macroscopic Analysis of Cardiac Morphology

Heart weights, normalized to tibia length, showed no significant difference between MFS mice and their WT littermates at the different ages studied (Figure 1). In addition, macroscopic analysis of the dissected hearts revealed no difference in cardiac dimensions (length and width) between the MFS and WT mice in any of the age groups (Appendix A
Figure A1).

### 2.3. Myocardial Structure

#### 2.3.1. Macroscopic Analysis of Ventricular Wall Morphology

Further macroscopic analysis of the hearts revealed the presence of myocardial tears (elongated dark red lines) and vesicles (dark red vesicular bulging lesion) at the level of the right ventricular (RV) surface of the MFS mice (Figure 2a–d). Both the tears and the vesicles had a dark red appearance, and the vesicles expanded slightly during post-mortem reflex contractions of the heart, suggesting a direct connection to the RV cavity. Interestingly, these lesions were not present in any of the WT mice, whereas the majority of the hearts of the MFS mice presented these lesions at each age group studied (Figure 2e).

#### 2.3.2. Histological Analysis of Ventricular Wall Lesions

In order to identify the nature of the lesions in the RV free wall, histological staining was performed on consecutive myocardial tissue sections in which a tear or vesicle was present (Figure 3). Standard HE staining revealed that both the tear as well as the outer wall of the vesicle (Figure 3a–d) are composed of a relatively thin layer of epicardial cells embedded in ECM components. The tear consists of a thinned region in the RV wall lacking cardiac muscle tissue adjacent to the epicardial layer. The vesicle-like lesions are connected with the lumen of the RV through a small duct between cardiac muscle lamellae (Figure 3c). Furthermore, in-depth cellular morphology analysis of the HE-stained tissue sections did not show any evidence of inflammatory cell infiltration at the site of or near the myocardial wall lesions (Figure 3b,d,e,g). Picrosirius red (PSR) staining (Figure 3f,h), as well as a fluorescent dye (Figure 3i–p), showed that both the tear and the vesicle wall contain abundant levels of collagen. The cellular border of these cardiac lesions consists mainly of fibroblasts, as indicated by a positive vimentin staining (Figure 3i–l), with an inner lining consisting of CD31-positive endothelial cells (Figure 3q,s). Adjacent to the myocardial tear, a discrete number of cells stained positive for the myofibroblast marker α-smooth muscle actin (α-SMA, ACTA2) (Figure 3n). Relative to the tear, the number of myofibroblasts is higher in the wall of the vesicle (Figure 3o,p), where a layer of subendothelial cells was shown to express α-SMA. No cardiac myocytes were located in the tear nor the vesicle wall, as evidenced by the absence of α-actinin-2 (ACTN-2) staining (Figure 3r,t, respectively).

#### 2.3.3. Myocardial Wall Structure

Micrographs of cross-sectioned myocardial tissue stained for fibrillin-1 showed an almost complete loss of fibrillin-1 staining in the sections of MFS mice (Figure 4a,b). Fibrillin-1 staining was abundantly present in the myocardial tissue sections of WT mice located between the muscle lamellae of the trabeculated myocardium and at the level of the subepicardial region, with strong fibrillin-1 staining along the RV free wall. In contrast, very limited fibrillin-1 staining was observed throughout the myocardial tissue section of the MFS mice. Analysis of HE-stained cardiac sections showed a prominent decrease in compaction of the ventricular myocardial tissue, demonstrated by more free space between the cardiac muscle lamellae (representative images in Figure 4c,d). The calculated index for cardiac non-compaction increased with age in MFS mice and was significantly higher compared to WT littermates at the age of 3 months (Figure 4e).

### 2.4. TGF-β-Dependent Signaling Pathways

Western-blot analysis was performed on LV myocardial tissue samples of 3-month-old WT and MFS mice in order to investigate the contribution of canonical (Smad2/3) and noncanonical (ERK1/2) TGFβ-dependent signaling in the myocardium. The ratio of phosphorylated Smad2, Smad3, and ERK to total respective proteins was calculated to document the activation of the specific pathway. Compared to WT littermates, the MFS mice did not show a significant difference in either Smad2, Smad3, or ERK activation (Mann–Whitney U-test; *p* ≥ 0.1111, Figure 5).

### 2.5. Electrocardiographic Analysis

#### 2.5.1. Analysis of 24 h Recordings

No significant difference could be observed between the average RR-peak intervals of the MFS mice and their WT littermates (respectively 102.1 ms ± 3.1 and 97.3 ms ± 0.9 ms, independent samples test; *p* = 0.195). In contrast, decreased heart rate variability (HRV) was observed in the MFS mice, indicated by a significantly lower full-width at half-maximum interval of the RR-peak interval distribution histogram compared to WT mice (respectively 11.3 ms ± 1.0 ms and 16.1 ms ± 1.0 ms, independent samples test; *p* = 0.014).

The occurrences of arrhythmic events over 24 h were counted and classified according to the scheme presented in Appendix B
Figure A2 (Table 2). Although arrhythmic events also occurred to a limited extent in WT mice, the number of arrhythmic events was significantly higher in MFS mice (Mann–Whitney U-test; *p* = 0.019). Next, the number of each type of arrhythmic event was calculated as a relative percentage of the total number of arrhythmic events per genotype. The type of arrhythmic event highly depended on the genotype (Pearson Chi-square test; *p* < 0.001) with the main arrhythmic event in WT mice being sinus arrest (48.7%), whereas in the MFS mice, extrasystoles were the most abundant (total of 96.3%).

#### 2.5.2. Analysis of Short-Term Heart Rate Variability

A total of 27 paired fragments of two minutes which did not contain arrhythmic events, obtained from the paired WT and MFS traces studied for long-term analysis, were selected for short-term HRV analysis. The standard deviation of normal to normal RR-intervals (SDNN), a frequently used parameter of HRV, was significantly lower in the MFS recordings compared to the WT recordings, on average 2.77 ms ± 0.25 ms versus 5.31 ms ± 0.42 ms, respectively (Table 3, Mann–Whitney U-test, *p* < 0.001). Furthermore, the frequency-domain (total power, high and low frequency (HF and LF, respectively)) parameters of HRV, obtained by means of power spectral density (PSD) analysis, were significantly reduced in the MFS mice compared to the WT mice. Total power decreased 3-fold and HF decreased 4-fold, while LF had a 10-fold decrease (Table 3, Mann–Whitney U-test; *p* < 0.001).

## 3. Discussion

Clinical management and surveillance of cardiovascular manifestations of MFS (with a particular focus on aortic dilatation and mitral valve prolapse) has improved significantly over the past decades, resulting in a higher life-expectancy of MFS patients [5,6,7]. This resulted in the appreciation of additional MFS-related cardiac manifestations, including intrinsic myocardial dysfunction and arrhythmia [8,9,10,11,12,13], which had previously remained undetected. MFS mouse models have already been used previously in order to study the MFS-related cardiac phenotype [11,21,22]. However, the number of studies is limited, especially in the context of MFS and arrhythmia.

We performed both in vivo and ex vivo in-depth cardiac phenotyping of the hypomorphic *Fbn1^mgR/mgR^* MFS mouse model. We first confirmed the presence of severe aortic dilation extending from the sinuses of Valsalva to the descending thoracic aorta in MFS mice, which recapitulates earlier findings in this mouse model [23]. Based on macroscopic evaluation of cardiac structure and ultrasound measurements of cardiac function, no significant difference was observed between the LV dimensions and systolic function of MFS mice compared to WT mice. On the other hand, we detected diastolic dysfunction, characterized by a significant decrease in E-wave velocity in the MFS mice compared to their WT littermates. This decrease in E-wave velocity, indicative of slower early filling of the LV, indicates that fibrillin-1 might play an important role in the active relaxation of the LV myocardium [2,24]. Similar findings have been observed previously in children with MFS [24,25].

In contrast to our current results, a previous study of the cardiac phenotype of the *Fbn1^mgR/mgR^* mouse model identified a severe dilatation of the LV associated with significant systolic dysfunction [22]. The discrepancy in cardiovascular manifestations between this study and ours might be explained by environmental factors and/or the genetic background of the mice. It is interesting to note that we used *Fbn1^mgR/mgR^* mice on a congenic C57BL/6J background whereas Cook et al. [22] studied a mixed 129 background ((129X1/SvJ × 129S1/Sv)F1-Kitl+) [22,23]. It has been demonstrated before that the genetic background of mice can have a significant influence on their cardiovascular physiology both at baseline and after induction of pathological stress [26,27,28,29,30,31]. Interestingly, it has previously been shown that the *Fbn1^mg^^ΔloxPneo/+^* hypomorphic mouse model of MFS also shows phenotypic variability associated with genetic background, as heterozygous mice showed a more severe aortic and skeletal phenotype on the 129/Sv background than on the C57BL/6 background [32]. Analogous to the approach used to map genetic modifiers of the aortic and skeletal manifestations in this *Fbn1^mg^^ΔloxPneo/+^* model [33], it will be interesting to perform a linkage analysis on *Fbn1^mgR/mgR^* mice on a mixed C57BL/6J/129 background in order to identify new modifiers involved in the disease mechanisms leading to the MFS-related cardiac manifestations. This could reveal mechanisms which might also be relevant to patients with MFS.

In order to begin to investigate the molecular mechanisms responsible for the observed cardiac manifestations, we assessed the level of activation of the canonical and non-canonical TGFβ-dependent signaling pathways. Similar to the results from Cook et al. [22], we found no significant changes in the level of Smad2/3 activation in the myocardial tissue of our *Fbn1^mgR/mgR^* mouse model. The level of ERK1/2 phosphorylation however was also unchanged in the myocardial samples of our MFS mice, while this pathway was significantly activated in the heart of the MFS mouse model used by Cook et al. [22] Based on these and other previously published results together with our current data, the level of activation of ERK1/2 via the non-canonical TGFβ signaling pathway seems to correlate with the development of LV dilatation and systolic dysfunction in MFS mice [11,21,22]. This pathway is therefore a likely candidate target of (a) strain-specific genetic modifier(s). Of note, the observation that our model shows a similar level of aortic dilatation as the model studied by Cook et al. [22], while both models have a markedly different cardiac phenotype, again, underscores the concept that fibrillin-1 defects lead to primary cardiac manifestations, independent of the aortic phenotype [11,22].

At the level of the RV free wall of the *Fbn1^mgR/mgR^* mice, we identified two types of previously unreported lesions. The outer layer covering the lesions had a similar composition as the epicardium (fibroblasts embedded in collagen), whereas the inner layer consisted of endothelial cells in contact with the RV lumen. A subset of fibroblasts stained positive for α-SMA expression, indicating they had switched to a myofibroblast phenotype. While only a limited number of myofibroblasts were found in the vicinity of the myocardial tear, a larger number were identified in the subendothelial layer of the vesicle wall. These findings suggest that the fibroblast lineage switch occurred progressively during development of the cardiac lesions, likely in response to the altered local mechanical tension. Myofibroblasts are known to be generated after injury and play a role in tissue remodeling by increasing collagen deposition and providing contractile force to the weakened tissue. We furthermore observed a lack of cardiomyocytes in the walls of both types of cardiac lesions. Based on these characteristics, the lesions can be classified as pseudoaneurysms of the RV free wall [34,35,36]. We hypothesize that the etiology of the RV pseudoaneurysms in the *Fbn1^mgR/mgR^* mouse model can be assigned to the reduced ability of the remaining fibrillin-1 fibers to maintain strong structural connections between adjacent myocardial muscle lamellae. The absence of similar pseudoaneurysm lesions in the LV wall is surprising considering the higher intraventricular pressures in this cardiac chamber. It is likely that the increased thickness of the LV myocardial wall, including a relatively fibrillin-1-poor outer compact myocardium [2], prevents the development of transmural lesions in this cardiac chamber. In humans, one case of a LV pseudoaneurysm in a patient with MFS has been reported which was related to apical venting at the occasion of aortic surgery [37]. The LV pseudoaneurysm was detected by means of ultrasound imaging and had a cavity width of 40 mm. Our study was performed in mice where, due to their limited size, we were only able to detect the pseudoaneurysms by means of histology.

To the best of our knowledge, RV pseudoaneurysm formation has not been reported before in patients with MFS, nor in any MFS mouse model. Nevertheless, a number of explanations can be put forward for the lack of similar observations in humans. Firstly, the mouse model used for this study is considered as a Marfan model with an early-onset, severe phenotype; it is conceivable that the myocardial lesions are part of this specific, severe phenotype. Secondly, detecting small lesions in a trabeculated right ventricular wall with conventional imaging techniques is particularly challenging and will require dedicated high-resolution modalities. Thirdly, these lesions may have remained undetected so far because autopsies are not routinely performed on deceased MFS patients, combined with the fact that severe (cardiovascular) forms of MFS in humans are rare [34,35,36]. With this research we hope to increase awareness for this new myocardial manifestation of MFS, which could potentially lead to new data regarding the prevalence of this type of cardiac lesions in MFS patients and other MFS mouse models. Similar to the variable presence of dilated cardiomyopathy in *Fbn1^mgR/mgR^* mice, it is likely that this novel cardiac manifestation is dependent on genetic modifiers associated with the genetic background, possibly explaining why this phenotype has not been observed previously in another study using mouse models of MFS. Further research is also warranted to investigate whether these ventricular pseudoaneurysms in a MFS mouse model are prone to rupture or increase the risk of arrhythmias and thromboembolism, as is known in the human setting of ventricular pseudoaneurysms [34,35].

Histological analysis of the myocardial tissue showed a highly reduced fibrillin-1 deposition associated with a decreased compaction of the myocardial tissue (resembling LV non-compaction cardiomyopathy (LVNC)), which deteriorated with age in MFS mice compared to WT mice. Interestingly, pathogenic variants in the *FBN1* gene [38,39] as well as pathologically increased activation of the TGFβ signaling pathway, of which the bioavailability is regulated by fibrillin-1 [40,41,42], have already been linked with LVNC. Under physiological conditions, fibrillin-1 is abundantly deposited at the level of the inner trabeculated myocardium [2,3], which is also the area showing the most prominent decrease in myocardial compaction in our MFS mouse model. Based on these findings, it is plausible that fibrillin-1 assures a tight connection between adjacent myocardial muscle lamellae, thereby preserving the compaction of myocardial tissue. A strongly reduced fibrillin-1 deposition in myocardial tissue, as observed in the MFS mice, might thus weaken this connection which deteriorates with age due to continuous high pressures and growth of cardiac tissue. Interestingly, the fibrillin fibers in the RV free wall appear to cross the entire myocardial wall in WT mice, while a significantly decreased compaction can also be appreciated in this area in the *Fbn1^mgR/mgR^* hearts. This is likely linked to the susceptibility of the RV wall to the development of the pseudoaneurysm lesions which we discovered in this model.

Finally, based on the observed rhythm disorders in MFS patients [14,16], we performed a detailed assessment of the cardiac rhythm phenotype in the *Fbn1^mgR/mgR^* mice. To the best of our knowledge, this is the first study to perform continuous 24 h ECG recording in a MFS mouse model. Our data revealed a decreased HRV in MFS mice compared to WT mice which is in accordance with published data on HRV in patients with MFS [43,44]. A decrease in HRV indicates a possible degree of autonomic dysfunction, which is associated with an increased risk for malignant ventricular arrhythmia [43,44,45,46]. This implies that a decreased HRV could be implemented as a prognostic marker for cardiac rhythm disorders in MFS patients. Furthermore, the frequency domains of HRV were analyzed by means of PSD which revealed a reduced absolute power of both the HF and LF bands, and a subsequent reduction in the total power in the MFS traces compared to the WT traces. This implies that both the parasympathetic (with HF as the indicator) and sympathetic (with LF as the indicator) activity on the sinoatrial node is diminished [47]. Unfortunately, the exact mechanism underlying MFS-related HRV suppression remains unknown. Further research regarding this subject on the MFS mouse model used in this study might potentially lead to new insights on the molecular mechanisms underlying this phenomenon. The analysis of the ECG recordings also revealed a higher prevalence of arrhythmic events in the MFS mice compared to their WT littermates, with a high frequency of extrasystoles. Extrasystoles have frequently been reported in patients with MFS [14,16]. Based on the ECG data, the *Fbn1^mgR/mgR^* mouse model clearly recapitulates the ECG abnormalities observed in patients with MFS, making it an interesting model for further research exploring the pathological mechanisms leading to the cardiac rhythm disorders associated with MFS. Interestingly, previous studies have shown that LVNC is associated with an increased occurrence of ventricular arrhythmia [39,48,49]. Since we have demonstrated a strong reduction in myocardial compaction in our MFS mouse model, we hypothesize that a lack of myocardial connectivity is directly linked to the increased prevalence of arrhythmic events observed in the *Fbn1^mgR/mgR^* mice.

## 4. Materials and Methods

### 4.1. Mice

International, national, and institutional guidelines for implementation of good animal welfare were adhered to. All animal procedures were approved by the local ethical committee of the Faculty of Medicine and Health Sciences at Ghent University (project identification codes ECD16/14 and ECD18/15, approved on 18 May 2016 and 23 April 2018, respectively) and were conducted in compliance with directive 2010/63/EU. Male congenic *Fbn1^mgR/mgR^* mice (originally designated as *B6.129-Fbn1^tm2Rmz^/J*, The Jackson laboratory, stock nr.005704) and WT littermates backcrossed for at least 10 generations on a C57BL/6J background were used for this study. The *Fbn1^mgR/mgR^* mice are referred to as “MFS mice” in this manuscript.

### 4.2. Ultrasound Analysis

Ultrasound imaging of the aorta and the heart was performed on 3-month-old MFS and WT mice (*n* = 12 per group) in order to obtain both functional and structural cardiovascular data. Prior to ultrasound analysis, mice were anesthetized (1.0–1.5% isoflurane, 0.5 L/min 100% O_2_) and placed in supine position on a heated platform connected to ECG electrodes. Subsequently, chest hair was removed with hair removal cream. Throughout the cardiovascular ultrasound recording, heart rate, respiration rate, and body temperature were continuously monitored. Recordings were performed with a Vevo 2100 ultrasound apparatus (Visualsonics, Toronto, ON, Canada) equipped with a high-frequency linear array transducer (MS 550D, frequency 22–55 MHz). Finally, the obtained ultrasound recordings were analyzed offline with the Vevo LAB software program (version 3.1.1, Visualsonics), whereby measurements were performed in triplicate. For the analysis of systolic and diastolic ventricular parameters, only measurements obtained at a heart rate ≥350 bpm were included. Measurements were done by a single investigator, blinded to the experimental group. Detailed materials and methods for the ultrasound analysis are provided in Appendix C.

### 4.3. Electrocardiographic Analysis

ECG traces of 24 consecutive hours were obtained in MFS mice and WT littermates of 10 weeks of age (*n* = 4 per group) according to the protocol described by Steijns et al. [50]. Four days post-operative data were stored for further offline analysis.

For long-term analysis of the ECG traces, the acquired ECG data were subdivided into fragments of twenty minutes and subsequently analyzed in Matlab^®^ (version R2018b, MathWorks, Natick, MA, USA). RR-peak interval (ms) and HRV (ms), represented by the full-width at half-maximum interval of the RR-peak interval distribution histogram during sinus rhythm, was calculated per fragment. Furthermore, arrhythmic events were identified semi-automatically based on deviating RR-peak intervals and subsequent manual evaluation of the ECG traces.

For short-term analysis of the ECG traces, only simultaneously acquired twenty-minute fragments of a MFS and a WT mouse were selected, based on similar RR-peak intervals. Next, R-peak selection and PSD analysis of the HRV were performed on the fragments with the online software platform PhysioZoo (version 1.2.0, Technion, Haifa, Israel) [51].

### 4.4. Tissue Isolation

The MFS and WT mice were sacrificed by means of CO_2_-inhalation using gradual fill (1.0 L/min) at the age of 1, 2, and 3 months. A thoracotomy was executed in order to isolate the heart, which was subsequently flushed with a cooled 30 mM KCl solution to induce cardioplegia. After KCl-flushing the total heart weight was measured and macroscopic pictures were taken using a Leica MZ6 microscope (Leica Microsystems, Wetzlar, Germany) with a DFC425 color camera (Leica Microsystems) in order to analyze the heart length and width. For further histological analysis, the hearts were flushed and fixed for 20 h at 4 °C with 4% paraformaldehyde. After paraformaldehyde fixation, the hearts were stored in a 70% ethanol solution until further processing. Furthermore, the left tibia was amputated, and soft tissue was removed in order to obtain the tibia length, which was used for normalization of the heart weights. Heart weight correlates with tibia length and is less prone to fluctuation as observed for body weight. Furthermore, due to bony overgrowth the body- and tibia-length of the MFS mice was higher compared to the WT mice. For protein analysis, the left ventricle of the heart was isolated and stored at –80 °C until further processing.

### 4.5. Histological Analysis

After fixation of the hearts, dehydration and paraffin embedding was performed using an automated Leica TP1020 tissue processor (Leica Biosystems, Wetzlar, Germany) according to standard histological protocol. Subsequently, five-µm-thick serial cross-sectional tissue sections at the level of the mid-ventricles were obtained (Microm HM355S, Thermo Fisher Scientific, Waltham, MA, USA), deparaffinized, and rehydrated before staining. Hematoxylin-eosin (HE) staining [52] and a picrosirius red (PSR) staining [53] (to stain collagen deposits) were performed according to standard protocols. Fibrillin-1, fibroblasts, endothelial cells, and cardiac myocytes were detected by means of immunofluorescence with the primary antibodies anti-fibrillin-1 (pAb 9543, gift from prof. dr. Lynn Y. Sakai [54], 1/100 dilution), anti-vimentin (3932P, Cell Signaling Technology (CST, Danvers, MA, USA), 1/200 dilution), anti-ACTA2 (staining α-SMA, CST, 1/100 dilution), anti-CD31 (Ab28364, Abcam (Cambridge, UK), 1/100 dilution), and anti-ACTN-2 (Ab9465, Abcam, 1/50 dilution), respectively. Collagen deposition was also detected by means of fluorescence after overnight incubation with a collagen hybridizing peptide (15µM, F-CHP, 3Helix Inc., Salt Lake City, UT, USA) subsequent to heat treatment of the tissue sections. Detailed materials and methods for the (immuno) fluorescence staining protocol are provided in Appendix C.

Sections were visualized with a Zeiss fluorescence microscope (Axio Observer. Z1, Zeiss, Oberkochen, Germany) coupled to an Axiocam camera (Zeiss). To obtain an overview image of the entire tissue section after HE and fibrillin-1 staining, stitching software (Tiles-Zeiss, ZEN pro 2012, Zeiss) was used.

### 4.6. Myocardial Compaction Calculation

In order to analyze the compaction level of the myocardial tissue at the level of the mid-ventricles in HE-stained tissue sections, a custom image analysis algorithm was developed using FIJI software (LOCI, University of Wisconsin, Madison, WI, USA) [55]. Importantly, as staining conditions influence the results of the analysis, comparisons between the calculated compaction levels were only done on tissue sections which were stained and imaged at the same time using identical conditions. Tissue sections were grouped by age, and the result was normalized to WT values. Each age group contained three WT and three MFS mice and calculations were done in triplicate. Detailed materials and methods for the calculation of the myocardial compaction are provided in Appendix C.

### 4.7. TGFβ-Dependent Signalling

Protein levels of the canonical (Smad2/3) and non-canonical (ERK1/2) TGFβ-dependent signalling pathway were analyzed by means of Western blotting. Tissue samples mixed in lysis buffer (RIPA, Sigma-Aldrich, Saint Louis, MO, USA) complemented with protease inhibitors (Complete protease inhibitor cocktail tablets, Roche, Basel, Switzerland) and phosphatase inhibitors (Cocktail II and III, Sigma-Aldrich) at a 30% (*w*/*v*) ratio were centrifuged at 20,000 g and 4 °C for 30 min. Supernatant was collected and the protein concentration was determined (Pierce BCA protein kit, Thermo Fisher Scientific). Samples were reduced by addition of dithriothreitol (Sigma-Aldrich) and incubation at 9 °C for 5 min. Next, 30 µg of the sample was run on a NuPage 4–12% Bis-Tris gel (Invitrogen, Carlsbad, CA, USA) together with a lane marker (Thermo Fisher Scientific), and the proteins were subsequently transferred onto a nitrocellulose (Invitrogen, for pERK1/2 blots) or PVDF membrane (Invitrogen, for pSmad2/3 blots). The membrane was blocked in 2% bovine serum albumin (Sigma-Aldrich, for pERK1/2) or 5% ECL Blocking Buffer (Sigma-Aldrich, for pSmad2/3) and incubated overnight at 4 °C with anti-phospho-p44/42 MAPK (pERK, 1/2000, CST), anti-phospho-Smad2 (pSmad2, 1/500, CST) or anti-phospho-Smad3 (pSmad3, 1/1000, CST). Next, the membrane was incubated with the secondary antibody, anti-rabbit IgG HRP-linked (1/20000, CST) for one hour at room temperature. After incubation with a luminol-based HRP substrate (Thermo Fisher Scientific), the membrane was scanned using the Chemidoc-it imaging system (UVP, Sopachem, Nazareth, Belgium). Next, the membrane was treated with stripping buffer (Thermo Fisher Scientific) and blocked with 2% bovine serum albumin (for total ERK1/2) or 5% ECL Blocking Buffer (for total Smad2/3) and the process was repeated with the primary antibody against p44/42 MAPK (total ERK, 1/1000, CST), Smad2 (1/1000, CST), or Smad3 (1/1000, CST). Image J software (version v. 1.44p, NIH, Bethesda, MD, USA) was used for quantification of the protein bands.

### 4.8. Statistics

Statistical analysis and generation of graphs were performed with the statistical software package SPSS (version 25.0, IBM, Armonk, NY, USA), where the acquired measurements were used as dependent variables and the genotype as independent variables. A Shapiro–WiIk test for normality was applied to the acquired numerical data. Normal-distributed variables were analyzed with an independent samples test. Otherwise, variables deviating from a normal distribution and categorical variables were compared by performing a Mann–Whitney U-test. Statistical analysis of the frequencies of type of arrhythmic events was performed with a Pearson Chi-Square test. All results are presented as the mean ± standard error (SE). A *p*-value < 0.05 was defined as statistically significant (two-sided).

## 5. Conclusions

This study aimed to contribute to a better understanding of the cardiovascular manifestations of MFS, using the genetically modified *Fbn1^mgR/mgR^* mouse strain as a validated animal model. We found that the *Fbn1^mgR/mgR^* mouse model not only recapitulates the aortic phenotype, but also shows a distinct cardiomyopathy phenotype characterized by myocardial non-compaction and impaired diastolic function. Interestingly, we observed lesions in the RV cardiac wall of MFS mice resembling pseudoaneurysms, which have not yet been reported in previous studies. Finally, we detected heart rhythm abnormalities in the MFS mouse model similar to those reported in patients with MFS, which manifested as a decrease in HRV and an increase in extrasystolic events compared to controls. The presence of RV pseudoaneurysms and heart rhythm abnormalities might be related to the decreased myocardial compaction that was observed in the *Fbn1^mgR/mgR^* mice. Taken together, our results expand the range of phenotypic manifestations of MFS in the heart, and might potentially contribute to improved diagnostics and better clinical management for patients with MFS.

## Figures and Tables

**Figure 1 ijms-21-07024-f001:**
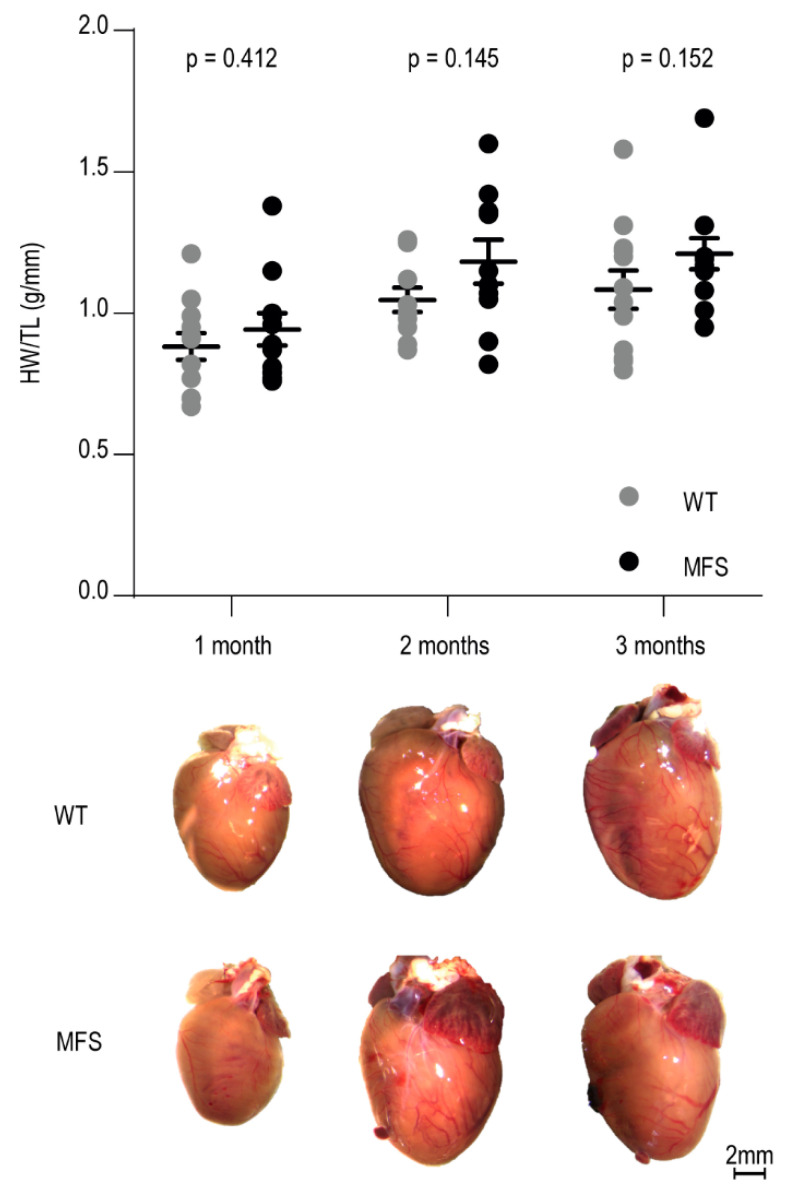
Macroscopic characterization of the heart of WT and MFS mice at 1, 2, and 3 months of age. Dot plot visualizes individual heart weight/tibia length ratios and mean value per age and genotype (*n* = 12 per age and genotype). WT and MFS measurements are indicated in grey and black, respectively. Error bars represent standard error. Associated *p*-values are annotated at the top of the graph (independent samples test on the mean values of the HW/TL). At the bottom of the figure, representative macroscopic pictures of WT and MFS mice hearts collected at 1, 2, and 3 months of age are depicted. Scale bar: 2 mm. HW/TL, heart weight to tibia length ratio (g/mm); WT, wild-type; MFS, Marfan syndrome.

**Figure 2 ijms-21-07024-f002:**
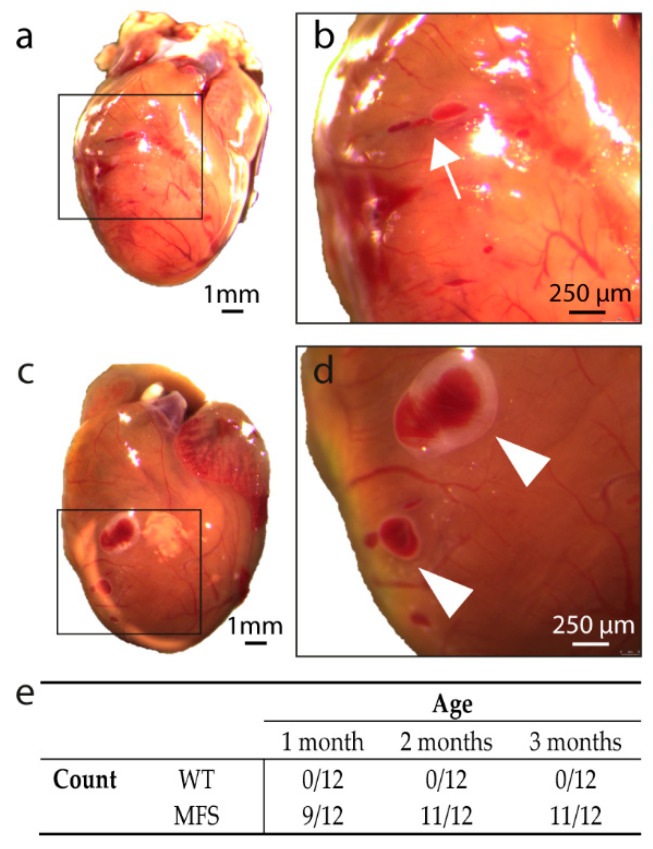
Macroscopic analysis of the heart of MFS mice. (**a**,**b**): macroscopic view of the heart of a MFS mouse presenting a tear at the level of the right ventricular wall. The tear is indicated with a white arrow in panel (**b**). (**c**,**d**): macroscopic view of the heart of a MFS mouse presenting vesicles at the level of the right ventricular wall. Vesicles are indicated with a white arrowhead in panel (**d**). (**e**): number of WT and MFS mice presenting cardiac lesions per age group. Scale bar of panels (**a**,**c**) = 1 mm, scale bar of panels (**b**,**d**) = 250 µm.

**Figure 3 ijms-21-07024-f003:**
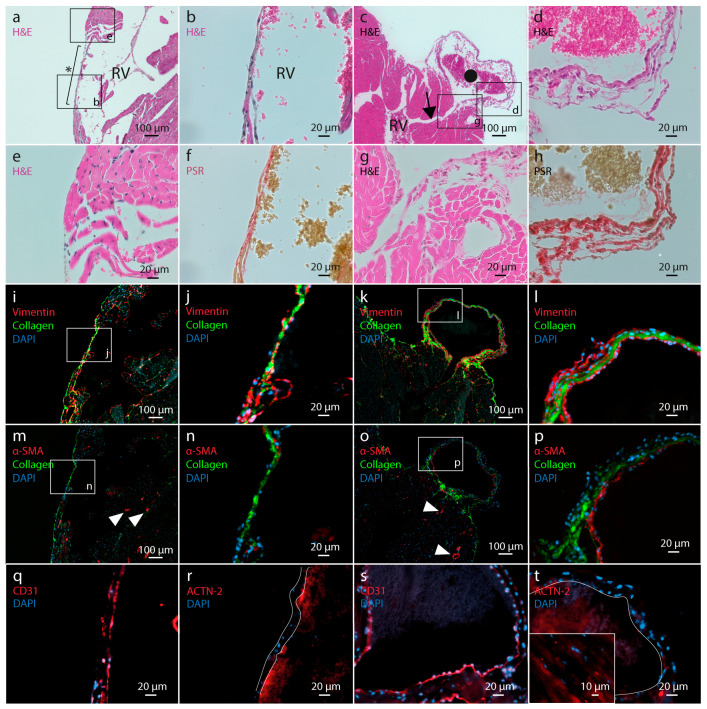
Microscopic analysis of tissue sections at the level of a myocardial tear and vesicle in the right ventricle free wall of the heart of MFS mice. (**a**,**c**): standard HE staining of representative myocardial tissue sections presenting a myocardial tear (indicated with *) and a vesicle (indicated with •). The black arrow in panel (**c**) indicates the narrow duct in between cardiac muscle lamellae connecting the lumen of the vesicle with the lumen of the right ventricle. Black boxes indicate localization of panels (**b**,**d**,**e**,**g**) within the tissue sections. (**b**,**d**,**e**,**g**): detailed images of HE staining of the myocardial lesions and the adjacent tissue. (**f**,**h**): picrosirius red staining (collagen deposition) of myocardial tissue sections at the level of a tear and vesicle, respectively. (**i**–**l**): double fluorescent staining of vimentin and collagen (stained red and green, respectively) in myocardial tissue sections presenting a myocardial tear and vesicle. Nuclei are stained with DAPI (blue)**.** (**m**–**p**): fluorescent staining of α-SMA (red) and collagen (green), with nuclei stained blue. Arrowheads indicate α-SMA staining of vascular smooth muscle cells in the blood vessel wall. (**q**,**s**): immunofluorescence staining of myocardial tissue sections at the level of a tear and vesicle, respectively, with an antibody directed against the endothelial marker CD31. CD31 positive cells are stained red. Nuclei are stained with DAPI (blue). (**r**,**t**): immunofluorescence staining of myocardial tissue sections at the level of a tear and vesicle, respectively, with an antibody directed against the cardiac myocyte marker ACTN-2. ACTN-2 positive cells are stained red. Nuclei are stained with DAPI (blue). Boundaries of the tear are indicated with white lines in panel (**r**). Vesicle wall is indicated with white line in panel (**t**). Insert in panel (**t**) represents positive ACTN-2 staining at the level of the Z-discs of the cardiac myocytes. HE, hematoxylin-eosin staining; PSR, picrosirius red staining; RV, right ventricle.

**Figure 4 ijms-21-07024-f004:**
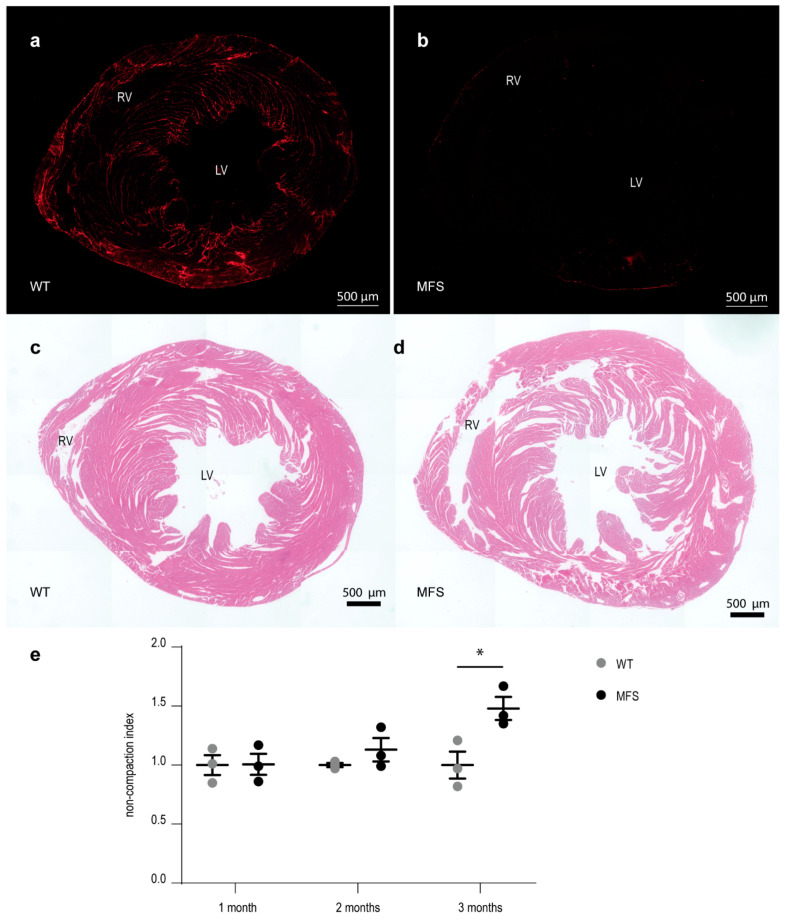
Decreased compaction of the ventricular myocardial tissue in MFS mice correlates with a decreased presence of fibrillin-1 staining. (**a**,**b**): overview picture of fibrillin-1 (red) staining of myocardial tissue sections of a 3-month-old WT and MFS mouse, respectively. (**c**,**d**): representative overview pictures of HE-stained myocardial tissue sections of a 3-month-old WT and MFS mouse, respectively. (**e**): graphical presentation of the myocardial compaction level through the measurement of the average non-compaction index of the myocardial tissue sections (*n*/age group = 3). Within each age group, values are normalized to the WT values. WT and MFS measurements are indicated in grey and black, respectively. Error bars represents standard error. * *p* = 0.032 (independent samples test). RV, right ventricle; LV, left ventricle; WT, wild-type; MFS, Marfan syndrome. Scale bar = 500 µm.

**Figure 5 ijms-21-07024-f005:**
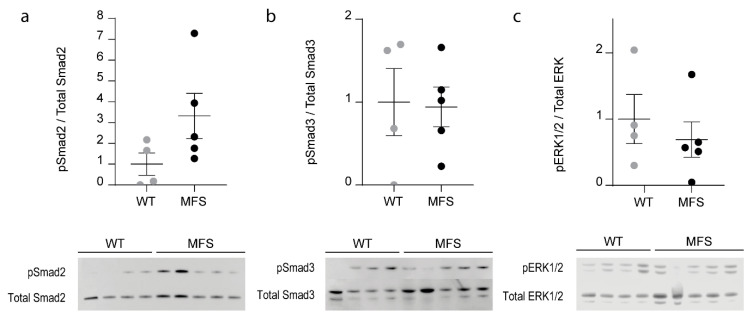
Canonical and non-canonical TGFβ-dependent signaling in the myocardial tissue of 3-month-old male WT and MFS mice (*n* = 4 and 5, respectively). (**a**): no significant difference could be observed in Smad2 activation (pSmad2/Total Smad2) between the MFS mice and their WT littermates (Mann–Whitney U test; *p* = 0.1111). (**b**): no significant difference could be observed in Smad3 activation (pSmad3/Total Smad3) between the MFS mice and their WT littermates (Mann–Whitney U test; *p* = 0.9048). (**c**): no difference in ERK activation (pERK/ERK ratio) could be observed between the MFS mice and their WT littermates (Mann–Whitney U test; *p* = 0.4127). Dot plots visualize normalized individual ratios and mean value per genotype. WT and MFS measurements are indicated in grey and black, respectively. Error bars represent standard error. WT, wild-type; MFS, Marfan syndrome.

**Table 1 ijms-21-07024-t001:** Physiological parameters of the aorta and cardiac ventricles of wild-type (WT) and Marfan syndrome (MFS) mice obtained by ultrasound measurements at 3 months of age. Measurements of the aorta include 12 WT and 12 MFS mice. Measurements of the cardiac ventricles include 5 WT and 6 MFS mice. *p*-values were calculated using an independent samples test.

	WT	MFS	*p*-Value
**Ao Diameters**			
**Ao Sinus (mm)**	1.61 ± 0.10	2.15 ± 0.09	0.001
**Ao Asc (mm)**	1.55 ± 0.10	2.89 ± 0.10	<0.001
**Arch (mm)**	1.43 ± 0.06	2.01 ± 0.06	<0.001
**Ao Desc (mm)**	1.15 ± 0.07	1.37 ± 0.06	0.038
**LVESD (mm)**	3.27 ± 0.08	3.24 ± 0.16	ns
**LVEDD (mm)**	4.20 ± 0.11	4.49 ± 0.17	ns
**RVEDD (mm)**	1.92 ± 0.10	2.11 ± 0.09	ns
**LV FS (%)**	22.14 ± 1.30	27.71 ± 2.88	ns
**E (mm/s)**	625.3 ± 54.2	310.5 ± 56.8	0.002
**A (mm/s)**	386.3 ± 36.5	382.0 ± 35.3	ns
**E/A**	1.67 ± 0.14	0.84 ± 0.14	0.002

WT, wild-type; MFS, Marfan syndrome; Ao, aorta; Ao Asc, ascending aorta; Ao Desc, descending aorta; LVESD, left ventricular end systolic diameter; LVEDD, left ventricular end diastolic diameter; RVEDD, right ventricular end diastolic diameter; LV FS, left ventricular fractional shortening; E, early left ventricular filling velocity; A, late left ventricular filling velocity; ns, not significant. Values are mean ± SE.

**Table 2 ijms-21-07024-t002:** Absolute total number and relative percentage of arrhythmic events per genotype over 24 h.

	Extrasystole			Sinus Arrest	Blocked p-Wave	AV-Block	Total
	Single	Doublet	≥3				
**WT ^A^**	2	0	0	14	12	3	31
**WT ^B^**	33	0	0	27	3	0	63
**WT ^C^**	6	0	0	29	2	0	37
**WT ^D^**	14	1	0	3	1	0	19
**Total Count WT** **(Relative %)**	55 (36.6%)	1 (0.7%)	0 (0.0%)	73 (48.7%)	18 (12.0%)	3 (2.0%)	150
**MFS ^A^**	50	4	0	3	7	0	64
**MFS ^B^**	894	275	2	9	1	0	1181
**MFS ^C^**	204	27	1	10	7	0	249
**MFS ^D^**	41	16	0	17	3	1	78
**Total Count MFS** **(Relative %)**	1189 (75.6%)	322 (20.5%)	3 (0.2%)	39 (2.5%)	18 (1.2%)	1 (0.0%)	1572
							*p* = 0.019

Mice are identified per genotype by means of ^A^, ^B^, ^C^ and ^D^; WT, wild-type; MFS, Marfan syndrome.

**Table 3 ijms-21-07024-t003:** Short-term analysis of heart rate variability. *n* = 27 paired electrocardiogram (ECG) fragments of two minutes obtained in 4 WT and 4 MFS mice.

	WT	MFS	*p*-Value
**SDNN (ms)**	5.31 ± 0.42	2.77 ± 0.25	<0.001
**PSD Analysis**			
**Total Power**	31.37 ± 5.93	8.98 ± 2.17	<0.001
**HF**	2.36 ± 0.33	0.55 ± 0.04	<0.001
**LF**	10.84 ± 1.82	1.06 ± 0.22	<0.001

WT, wild-type; MFS, Marfan syndrome; SDNN, standard deviation of normal R-intervals; PSD, power spectral density; HF, high frequency; LF, low frequency. Values are mean ± SE.

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
