# Peer review of "Spontaneous Right Ventricular Pseudoaneurysms and Increased Arrhythmogenicity in a Mouse Model of Marfan Syndrome"

_ijms, 2020, doi:10.3390/ijms21197024_

Round 1
Reviewer 1 Report
Comments for the Authors:
In the manuscript titled ‘ Spontaneous right ventricular pseudoaneurysms and 2 increased arrhythmogenicity in a mouse model of Marfan syndrome’ Steijns et al., have meticulously characterized the myocardial phenotype and function in Fbn1mgR/mgR mice, with a severely reduced expression of fibrillin-1 and manifestation of Marfan syndrome (MFS). The manuscript is very well written in view of recent developments in this field of research and contributes substantially to advance the understanding of MFS in murine models. I congratulate the authors on such a well planned and executed investigation. Nevertheless I expect that the readers will really appreciate further insights on the following aspects which in my opinion will improve the quality of the manuscript even more-
- In the histological analysis of ventricular wall sections the investigators might also check the aspect of cellular infiltration particularly inflammatory cells at the sites of tear and vesicle wall. This may primarily be achieved in HE stained sections and subsequently by staining with cell lineage specific markers.
- It would also be insightful to provide an estimate about the plasma levels of circulating TGFβ in these mice with respect to wild type.
- TGFβ indeed leads to ERK activation downstream of Ras and the authors have shown that this pathway is potentially unaltered in the Fbn1mgR/mgR mice. However, there are other possibilities downstream of TGFβ-receptor ligation like the phosphorylation of stress kinase p38MAPK downstream of TAK1, and ROCK downstream of RhoA. Besides phosphorylation status of the Smad proteins will add considerable information of great value to the manuscript.
Reviewer 2 Report
Steijns and colleagues described in this paper post-mortem cardiac features of the mutant Marfan syndrome (MFS) mice Fbn1mgR/mgR, highlighting spontaneous pseudo-aneurysmal myocardial lesions at the level of the right ventricular free wall. In vivo echocardiographic analysis showed a decreased heart rate variability and increased prevalence of extrasystolic arrhythmic events, suggesting a novel cardiac phenotype in the Fbn1mgR/mgR MFS mouse model in terms of cardiac dysfunction and rhythm disorders, caused by fibrillin-1 deficiency.
However, there are some issues and major revisions that must be addressed:
- May these observation of severe pathological murine model with an early onset find some corresponding cardiac feature in MFS patients?
- While LVNC condition has been previously reported in MFS patients, I have some issue on heart lesions. If they ever showed it, how many MFS patients have developed heart lesions as those indicated in the paper? In case of human heart lesions in MFS patients, which type of lesion they showed? Has no one, nowadays, reported pseudo-aneurysmal lesion in echocardiographic or in post mortem analyses of MFS patients?
- Myocardial compaction calculation: is it a brand new method or have you used a previous measure of the same feature? Has it already been calculated the compaction index in post mortem analysis of MFS subjects? The method used for this measurement included analysis of haematoxylin-eosin
- Alpha-2 actinin antibody, described in the methods and adopted for Figure X panel, is able to very well detect the z-discs of sarcomeric structures. Unfortunately, in these figures the typical banded sarcomeric shape of cardiomyocytes is not fully appreciable. Please, replace the indicated panels with more evocative pictures.
- In order to better show the lesion architecture, specifically in case of vescicles, I suggest to perform a double staining on the same heart tissue section, by marking the collagen (with a specific antibody and not by a chemical staining) together with vimentin of, better, alpha-SMA (secretive fibroblast). Alternatively, in order to dissect the specific cell role of fibroblasts into the lesions and the absence of cardiomyocytes, a double immunofluorence staining marking vimentin (or alpha-SMA) and alpha-2 actinin is suggested.
- Why only the pERK/ERK western blot has been tested, and not the canonical SMAD TGF-β pathway?
Round 2
Reviewer 1 Report
The investigators have addressed all the issues raised during the previous review process adequately. The revised manuscript furnishes new data to support their obserrvations. I congratulate the authors on having created an elegnt manuscript and hope that it will be received with great interest from the readers.
Author Response
We would like to thank the reviewer for their efforts in reviewing our manuscript and for their very positive assessment of our work.
Reviewer 2 Report
In the opinion of this Reviewer, the issues raised in the previous revision step have been almost completely addressed. However, there are some issues that require minor revisions.
- Concerning the correspondence between the Marfan model cardiac phenotype and human clinical manifestations, there are some points that must be better clarified. In the rebuttal letter, the authors first commented that the relevance of observations in animal models depends on the level of correlation with human conditions, and there are many examples where the translation of observations in mice has not been achieved in a human context. Furthermore, the authors declare that "the histological myocardial abnormalities observed in this mouse model have not yet been found and described in humans". Furthermore, it is presented that the Fbn1mgR/mgR model is a unique model, with an early and more severe phenotype than that presented in the "classical Marfan syndrome". Therefore, since it is known that in Marfan patients "myocardial dysfunction is generally mild and that severe disease appears to be limited to a small subgroup of patients" it would be correct to more strongly specify this passage in the paper discussion, highlighting the current inconsistency between the mouse model and human pathology. However, this Reviewer agrees that this study is certainly useful to complete the characterization of the Marfan Fbn1mgR/mgR mouse with a C57BL/6J background. Nonetheless, it would be better to emphasize that a human cardiac correspondence may not find, given also the difficulty of finding Marfan patients’ autopsy samples.
- Although the case reported from Janssen et al. (doi:10.1093/ejechocard/jen225) describes a bilobar apic pseudoaneurysm of a Marfan Syndrome patient due to left ventricular venting procedure, it could be useful to cite it in the revised version of your manuscript. This, in fact, may better help to distinguish differences between a human pseudoaneurysm, appreciable by echocardiography, and micro alterations as those reported in your study, clearly visible only by histology.
- This Reviewer agrees with the adoption of an automated-based method for myocardial compaction calculation, especially if this index may pave the way for further studies on myocardial manifestation in Marfan patients. Concerning the correlation analysis provided, this Reviewer has some issue that requires explanations:
- Firstly, it is not clear what dots in the graph are representing: are dots the averages of repeated measurements in the same murine myocardial tissue? In this case, error bars are missing.
- Where are the values represented in the graph in the paper? The non-compaction index values in Figure 4e are normalized on 1 month of WT?
- If every dot represents a single measure in a single animal (e. n=3 WT vs n=3 MFS), it is important to highlight that, if we distinguish WT from MFS mice, we can obtain two different correlations. The first correlation concerns WT mice, with an R2 of approximately 0.96, confirming a strong agreement in the measures of the two methods. On the contrary, by analysing the correlation only in MFS mice, the R2 is 0.51 circa, suggesting a strong inaccuracy of manual measurement in comparison with automated method. Thus, be aware, because only the correlation in WT strongly confirm the higher accuracy of automated method in comparison with manual method.
